

# Serum immunoglobulin M is associated with the severity of coronary artery disease in adults

Yanan Zhang[1],*, Xi Qi[2],*, Siming Wang[2], Wenduo Zhang[2],
Ruiyue Yang[2], Xinyue Wang[2], Wenxiang Chen[2], Fusui Ji[2], Jun Dong[2]
and Xue Yu[2]

[1] The Affiliated Hospital of Qingdao University, Qing Dao, Shan Dong, China
[2] Beijing Hospital, Beijing, China
* These authors contributed equally to this work.

Corresponding author
Xue Yu, yuxuemd@aliyun.com

## ABSTRACT

**Purpose:** The purpose of this study was to investigate the relationship between serum immunoglobulin M (IgM) and the severity of coronary artery disease in Chinese patients who underwent coronary angiography.

**Methods:** A total of 2,045 patients who underwent coronary angiography (CAG) from March 2017 to March 2020 at Beijing Hospital were included in this study. Serum IgM concentration and biochemical indicators were measured before coronary angiography (CAG). The triquartile IgM levels at baseline in the population were analysed. Spearman rank correlation was used to analyse the association between IgM and traditional risk factors for coronary artery disease (CAD). CAD patients were divided into subgroups by affected area, number of affected vessels, and Gensini score to analyse the relationship between IgM and CAD severity. Multivariable logistic regression analysis was used to evaluate the association between IgM and CAD severity.

**Results:** Serum IgM levels were significantly lower in the CAD group (63.5 mg/dL) than in the non-coronary artery disease (NCAD) group (72.3 mg/dL) ($P < 0.001$). Serum IgM levels were significantly associated with sex. Serum IgM levels were positively correlated with traditional CAD risk factors such as TG, TC and LDL-C ($P < 0.05$), and negatively associated with the number of obstructed vessels, the number of affected areas, and Gensini scores. After adjusting for age, sex, smoking status, hypertension, dyslipidaemia, diabetes, stroke, and statin use history, a high IgM level was independently negatively associated with the severity of CAD expressed by the Gensini score.

**Conclusion:** We determined that serum IgM was independently negatively associated with the severity of CAD diagnosed by angiography in Chinese adults.

## INTRODUCTION

Coronary artery disease (CAD) is one of the most common chronic diseases and remains the leading cause of death worldwide (*Laslett et al., 2012*). Atherosclerosis, the primary

underlying factor for CAD, is a chronic immune inflammatory condition that is characterized by the formation of plaques in the arteries, along with the accumulation of lipids and cells, such as leukocytes, endothelial cells, and foam cells, in intimal layers (*Drosos, Tavridou & Kolios, 2015*). Since atherosclerosis is almost irreversible, early identification, intervention of risk factors, and the identification of new biomarkers are integral to the prevention and treatment of CAD (*Gilstrap & Wang, 2012*).

Recently, numerous studies have shown that activation of the immune system is associated with the occurrence and development of CAD (*Okabe et al., 2005*; *Tsiantoulas et al., 2014*; *Zhou et al., 2001*). Inflammation and immune function play important roles in the development and progression of atherosclerosis, as well as plaque rupture and thrombosis (*Tsiantoulas et al., 2014*). A recent network-driven integrative analysis of data from genome-wide association studies identified that B-cell immune responses play a causative role in CAD (*Huan et al., 2013*). Generally, B cells are divided into innate-like B1 cells and B2 cells. Immunoglobulin M (IgM), mainly produced by B1 cells (*Choi et al., 2012*), is the largest type of immunoglobulin in the human body. It is the first isotype produced prior to class switching (*Fellah et al., 1992*). IgM effectively recognizes and eliminates pathogens in the early stage of immune defense (*Boes, 2000*). It is mainly found in the blood and has complement activation, sterilization, agglutination, cytotoxic and cytolytic activity, and immune regulation functions (*Ehrenstein & Notley, 2010*). Recent animal experiments and human studies have demonstrated the atheroprotective role of IgM. A previous study showed that the level of IgM antibodies against phosphorylcholine (IgM anti-PC) was negatively associated with the atherosclerotic disease burden in patients in the acute phase of ST-elevation myocardial infarction (STEMI) (*Knudsen et al., 2019*). However, the study was limited to the relationship between IgM anti-PC and atherosclerosis in STEMI patients. Currently, there is insufficient evidence of an association between IgM and CAD severity in humans. Our study aimed to investigate the relationship between IgM and CAD severity in Chinese patients undergoing coronary angiography (CAG).

## METHODS

### Study population

The patients in this study were from Beijing Hospital Atherosclerosis Study (BHAS, ClinicalTrials.gov registration number NCT 03072797) and it was a cross-sectional study. A total of 2,970 hospitalized patients who were suspected of having CAD and had indications for CAG were admitted to Beijing Hospital from March 2017 to March 2020. The exclusion criteria were as follows: patients who had severe congenital heart disease or severe cardiac insufficiency (NYHA class IV and LVEF ≤ 20%), primary pulmonary hypertension, severe hepatic and renal dysfunction, and severe peripheral arterial disease or related conditions that were contraindications of cardiac catheterization or organ transplant; patients who were receiving radiotherapy or chemotherapy; patients who were pregnant or nursing; patients with substance use disorders; patients undergoing treatment for mental illness; and patients with missing IgM values. Obtain demographic and medical history data from hospital records. This study was approved by the Ethics Committee of

Beijing Hospital (2016BJYYEC-121-02), and all patients signed written informed consent forms.

## Coronary angiography

CAG was performed by an experienced interventional physician. All targeted coronary lesions in enrolled patients were analyzed using QCA software in the Allura Xper FD20 angiography system (Philips Healthcare, Best, the Netherlands) and with reference to the American Heart Association (AHA) classification. The coronary artery segments, degree of stenosis, and number of obstructed vessels were carefully confirmed by two cardiologists. According to the diagnostic criteria of the ACC/AHA, CAD was defined as the presence of at least 50% stenosis in any one of the coronary arteries (left main artery, left anterior descending artery, left circumflex branch, or right coronary artery) or major branches. If this level of stenosis is not present, it is classified as non-CAD (*Scanlon et al., 1999*). The severity of coronary stenosis in patients was estimated by the Gensini score, which was calculated using the stenosis degree and corresponding weight coefficient (*Gensini, 1983*). Each lesion was assigned a score of 0–32, which was determined by multiplying the percentage of stenosis by the coefficient defined for each major coronary artery and segment. The Gensini score of each patient was obtained by summing the calculated results. Among all patients undergoing CAG, 923 patients who had previous percutaneous coronary interventions (PCIs) were excluded from the calculation of Gensini scores 2,045 subjects were included in the final analysis.

## Laboratory assays

The serum IgM levels of all patients were measured before their CAG using an IgM assay kit (batch number: YZB/USA 4923-2014) on a Beckman Coulter AU640 automatic biochemical analyser. Serum concentrations of total cholesterol (TC), triglycerides (TG), high-density lipoprotein cholesterol (HDL-C), low-density lipoprotein cholesterol (LDL-C), fasting blood glucose (FBG), creatinine (Crea), alanine aminotransferase (ALT), aspartate aminotransferase (AST), total protein (TP) and albumin (ALB) were measured using assay kits from Sekisui Medical Technologies (Osaka, Japan) on a Hitachi 7180 chemistry analyser. Routine blood indices were tested using an automatic haematology analyser (Sysmex, Kobe, Japan). There is no missing data in our study.

## Statistical analysis

Data was analysed with SPSS 27.0 statistical software. Continuous variables were expressed as the mean ± standard deviation (SD) if the distribution was normal. If the distribution was skewed, the variables were expressed as median and interquartile range (IQR). Count data was reported as frequencies and percentages. Categorical variables were presented as both absolute (number of patients) and relative frequencies (percentage). The one-way ANOVA was conducted to compute the differences between continuous variables, while the non-parametric Kruskal-Wallis test was used to determine the differences when the data was not normally distributed. The Chi-square test was used for categorical variables. Spearman's correlation analysis was used to examine the associations between IgM and

traditional CAD risk factors. The relationship between IgM and Gensini scores was evaluated using multivariable logistic regression analysis. Potential confounding variables (age, sex, smoking status, obese or overweight, hypertension, dyslipidaemia, diabetes mellitus, history of stroke, and statin use history) were controlled for in the regression models. Multivariate logistic regression analysis was used to evaluate the relationship between IgM and Gensini scores in subgroups. Logistic regression was used to further analyse the interaction between IgM and other factors (age, sex, smoking status, obese or overweight, hypertension, dyslipidaemia, diabetes mellitus, history of stroke, and statin use history). Odds ratios (ORs) for high *vs*. low Gensini scores were estimated with the corresponding 95% confidence intervals (CIs). A two-sided test was applied, and a *P* value of less than 0.05 indicated a statistically significant difference.

## RESULTS

### Demographic and clinical characteristics of the subjects

A total of 2,045 hospitalized patients were included in our study. Baseline characteristics according to tertile IgM levels are summarized in Table 1. The results showed that individuals with higher IgM levels also had higher levels of TC, LDL-C, ALT and TP. Patients with higher IgM levels tended to be younger ($P < 0.05$) and predominantly female. The increase in IgM levels corresponded with a significant decrease ($P < 0.05$) in FBG, Crea, and Gensini scores. Furthermore, when IgM levels increased, the prevalence of hypertension, diabetes mellitus (DM) and dyslipidaemia significantly decreased ($P < 0.05$). The median serum IgM levels in the CAD group and non-coronary artery disease (NCAD) group were 63.5 (43.2–91.9) and 72.3 (49.9–102.0) mg/dL, respectively. IgM levels in the CAD group were significantly lower than those in the NCAD group ($P < 0.001$).

### Correlations between IgM and conventional CAD risk factors and CAD severity

Spearman correlation analysis showed that serum IgM levels were significantly positively associated with TG, TC, LDL-C, HDL-C, TP, and ALB ($P < 0.05$). In addition, serum IgM levels were significantly higher in females than in males. Serum IgM levels were also significantly negatively correlated with body mass index (BMI), FBG, Crea, WBC, the number of obstructed vessels, and the Gensini score ($P < 0.05$). However, an association between IgM and age, SBP, DBP, LYMP, or hs-CRP was not significant. The results are shown in Table 2.

### Associations between IgM and the severity of CAD

To analyse the relationship between IgM level and CAD severity, patients with CAD were classified into subgroups based on the number of regions with stenosis: 1–3 and >3 stenosed regions. IgM levels exhibited a significant decrease, and there was an increase in the number of stenosed regions ($P_{trend} < 0.05$). Then, the patients were divided into subgroups with one stenosed or >1 stenosed vessel. As shown in Fig. 1, the increase in the number of obstructed vessels ($P_{trend} < 0.001$) corresponded with a significant decrease in

**Table 1 Comparison of baseline characteristics of study population according to serum IgM tertile.**

| Characteristic[a] | Tertile of serum level of IgM (mg/dL) | | | |
| --- | --- | --- | --- | --- |
| | Low | Intermediate | High | Trend $p$-value |
| $N$ | 685 | 679 | 681 | – |
| Age, year | 65.7 ± 10.9 | 64.2 ± 10.8 | 64.6 ± 11.0 | 0.027 |
| Male , $n$ (%) | 469 (68.6) | 402 (59.1) | 325 (47.7) | <0.001 |
| BMI, Kg/m2 | 26.3 ± 6.7 | 26.4 ± 9.8 | 25.7 ± 5.6 | 0.147 |
| SBP, mmHg | 137.8 ± 17.6 | 136.7 ± 18.1 | 137.6 ± 19.2 | 0.507 |
| DBP, mmHg | 79.1 ± 11.3 | 79.4 ± 11.3 | 79.1 ± 11.9 | 0.893 |
| **Smoking status, $n$ (%)** | | | | |
| Never | 358 (52.3) | 354 (52.1) | 426 (62.6) | 0.002 |
| Former | 92 (13.4) | 94 (13.8) | 69 (10.1) | |
| Current | 231 (33.7) | 227 (33.4) | 184 (27.0) | |
| Hypertension, n (%) | 554 (80.9) | 508 (74.8) | 522 (76.7) | 0.023 |
| Diabetes, $n$ (%) | 324 (47.3) | 326 (48.0) | 270 (39.6) | 0.003 |
| Hyperlipidemia, $n$ (%) | 275 (40.1) | 291 (42.9) | 239 (35.1) | 0.023 |
| History of stroke, $n$ (%) | 83 (12.1) | 78 (11.5) | 54 (7.9) | 0.051 |
| **Statins use, $n$ (%)** | | | | |
| No | 417 (60.9) | 411 (60.5) | 466 (68.4) | 0.002 |
| Take statins intermittently | 72 (10.5) | 70 (10.3) | 64 (9.4) | |
| Take statins continuously over 1 year, $n$ (%) | 149 (21.8) | 140 (20.6) | 93 (13.7) | |
| FBG, mmol/L | 6.0 (5.2–7.7) | 6.0 (5.1–7.5) | 5.7 (5.0–7.3) | 0.008 |
| TC, mmol/L | 3.5 (3.1–4.1) | 3.7 (3.1–4.2) | 3.7 (3.2–4.4) | <0.001 |
| TG, mmol/L | 1.5 (1.1–2.1) | 1.5 (1.1–2.1) | 1.6 (1.2–2.2) | 0.076 |
| LDL-C, mmol/L | 2.0 (1.6–2.5) | 2.1 (1.6–2.6) | 2.1 (1.7–2.7) | 0.007 |
| HDL-C, mmol/L | 1.0 (0.8–1.2) | 1.0 (0.8–1.2) | 1.0 (0.8–1.2) | 0.126 |
| Crea, umol/L | 78.0 (69.0–87.8) | 76.0 (66.0–87.0) | 74.0 (64.0–86.0) | <0.001 |
| ALT, U/L | 19.7 (15.0–27.6) | 19.5 (14.8–27.8) | 18.1 (14.2–26.5) | 0.047 |
| AST, U/L | 20.5 (17.5–25.5) | 20.2 (17.4–24.8) | 20.1 (17.6–24.5) | 0.488 |
| TP, g/L | 64.5 ± 5.2 | 65.1 ± 5.1 | 66.0 ± 5.2 | <0.001 |
| ALB, g/L | 40.5 ± 3.0 | 40.7 ± 3.0 | 40.7 ± 3.0 | 0.151 |
| WBC, $10^9$ | 6.2 (5.3–7.4) | 6.1 (5.1–7.3) | 6.1 (5.1–7.3) | 0.062 |
| LYMPH, $10^9$ | 1.7 (1.4–2.2) | 1.8 (1.4–2.2) | 1.8 (1.4–2.3) | 0.396 |
| hsCRP, mg/dL | 0.2 (0.2–0.5) | 0.3 (0.2–0.5) | 0.3 (0.2–0.5) | 0.872 |
| IgM, mg/dL | 39.1 (32.2–45.0) | 65.6 (58.2–74.3) | 109.8 (93.3–137.7) | <0.001 |
| Gensini score | 15.0 (3.3–45.0) | 11.5 (1.5–36.0) | 8.0 (1.5–32.3) | <0.001 |

Notes:
Abbreviations: BMI, body mass index; SBP, Systolic blood pressure; DBP, Diastolic blood pressure; FBG, fasting blood glucose; TC, total cholesterol ; TG, triglycerides; HDL-C, high-density lipoprotein cholesterol; LDL-C, low-density lipoprotein cholesterol; Crea, creatinine; ALT, alanine aminotransferase; AST, aspartate aminotransferase; TP, total protein; ALB, albumin; WBC, white blood cell; LYMPH, lymphocyte; hsCRP, hypersensitive C-reactive protein; IgM, immunoglobulin M.
[a] Data are mean ± SD, median (interquartile range) for continuous variables, or percentage for categorical variables.

IgM levels. Additionally, the patients were further divided into subgroups according to tertiles of the Gensini scores. Similarly, the results showed that IgM gradually decreased as Gensini scores increased from tertile 1 to tertile 3 ($P_{\text{trend}} < 0.05$).

**Table 2 Correlations between IgM with conventional CAD risk factors and CAD severity.**

|  | Correlation coefficients | *P* |
| --- | --- | --- |
| Age | −0.042 | 0.055 |
| Gender | 0.191 | <0.001 |
| BMI | −0.070 | 0.002 |
| SBP | −0.025 | 0.252 |
| DBP | −0.010 | 0.660 |
| FBG | −0.062 | 0.006 |
| TG | 0.048 | 0.029 |
| TC | 0.101 | <0.001 |
| LDL-C | 0.071 | 0.001 |
| HDL-C | 0.048 | 0.028 |
| Crea | −0.097 | <0.001 |
| TP | 0.141 | <0.001 |
| ALB | 0.044 | 0.048 |
| WBC | −0.060 | 0.007 |
| LYMP | 0.025 | 0.259 |
| hsCRP | 0.001 | 0.970 |
| Obstrutive vessels | −0.132 | <0.001 |
| Gensini score | −0.112 | <0.001 |

**Note:**
Abbreviations: SBP, Systolic blood pressure; DBP, Diastolic blood pressure; FBG, fasting blood glucose; TC, total cholesterol; TG, triglyceride; HDL-C, high-density lipoprotein cholesterol; LDL-C, low-density lipoprotein chole sterol; Crea, creatinine; TP, total protein; ALB, albumin; WBC, white blood cell; LYMPH, lymphocyte; hsCRP, hypersensitive C-reactive protein.

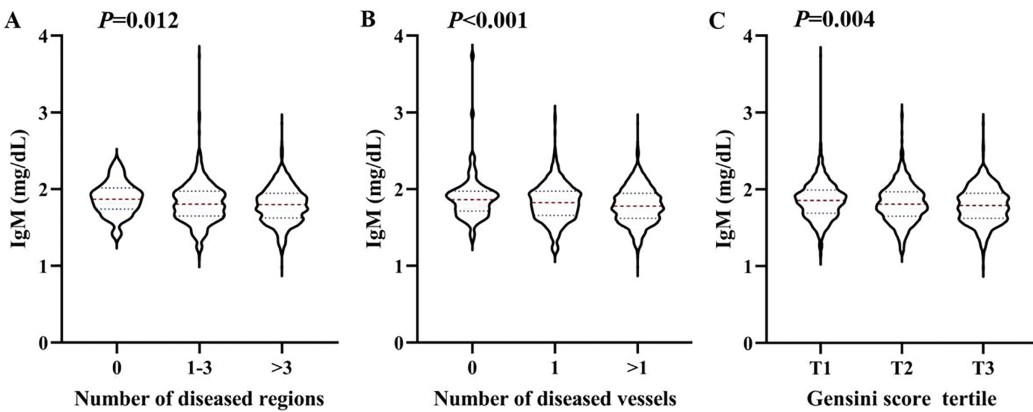

**Figure 1 Association between the serum IgM level and the severity of coronary artery lesion.** Violin plots of serum IgM concentrations at presentation with the number of stenosed regions (A), the number of stenosed vessels (B), and tertile of the Gensini scores (C), showing median (red dashed line) and interquartile ranges (blue dashed line) on a log 10 scale. *P* < 0.05 considered statistically significant.

**Table 3  Logistic regression analysis association between IgM and Gensini score.**

| | | P-value | OR (95% confidence intervals) |
|---|---|---|---|
| Model 1[a] | IgM | <0.001 | 0.687 [0.577–0.818] |
| Model 2[b] | Age | <0.001 | 1.464 [1.220–1.756] |
| | Gender | <0.001 | 0.448 [0.372–0.540] |
| | IgM | 0.006 | 0.776 [0.649–0.929] |
| Model 3[c] | Age | <0.001 | 1.499 [1.234–1.820] |
| | Gender | <0.001 | 0.511 [0.414–0.631] |
| | Smoking status | | |
| | Never | – | – |
| | Former | 0.183 | 2.596 [0.637–10.586] |
| | Current | 0.061 | 3.827 [0.938–15.619] |
| | Hypertension | 0.617 | 0.944 [0.754–1.182] |
| | Dyslipidemia | 0.616 | 0.944 [0.754–1.182] |
| | Diabetes | <0.001 | 2.618 [2.172–3.156] |
| | History of stroke | 0.416 | 1.119 [0.830–1.510] |
| | Statins application history | 0.833 | 1.025 [0.816–1.288] |
| | IgM | 0.017 | 0.798 [0.663–0.961] |

Notes:
[a] Model 1: Crude risk.
[b] Model 2: Adjusted for age and gender.
[c] Model 3: Further adjusted for smoking status, hypertension, dyslipidemia, diabetes, stroke, and statins application history. For gender, smoking status, hypertension, dyslipidemia, diabetes, history of stroke, statins application history, the references are: man, never smoked, no hypertension, no dyslipidemia, no diabetes, no history of stroke, and no statins application history, respectively.

## Logistic regression analysis of the association between IgM and CAD severity

We used logistic regression analysis to evaluate the association between IgM and Gensini scores. As shown in Table 3, the odds ratio (OR) decreased with increasing IgM levels in Model 1 ($P < 0.001$). High IgM levels were negatively associated with Gensini scores after adjustment for age and sex (Model 2) (OR = 0.776, 95% confidence interval (CI) [0.649–0.929], P < 0.01). A lower IgM level continued to be independently associated with Gensini score, which further adjusted for smoking status, hypertension, dyslipidaemia, diabetes, stroke, and statin use history (Model 3) (OR = 0.798, 95% CI [0.663–0.961], $P = 0.017$). Age and DM were positively associated with the Gensini score, whereas female sex was a protective factor for CAD, with an OR of 0.511 (0.414–0.631).

## Stratified analysis of the IgM level with CAD

Stratified analysis of the association between serum IgM level and the severity of coronary artery lesions was conducted in different age, sex, BMI, smoking status, hypertension, dyslipidaemia, DM, history of stroke, and statin use history subgroups. The effect of interactions between IgM levels and these factors on CAD severity was analysed. As shown in Fig. 2, there was a negative association between IgM levels and CAD severity among people younger than 70 years of age, with a BMI greater than 24 kg/m$^2$, who had never

| Subgroups | OR(95%CI) | | P for interaction |
|---|---|---|---|
| **Age** | | | |
| <70y | 0.730 (0.579–0.921) | | |
| ≥70y | 0.835 (0.608–1.147) | | 0.029 |
| **Gender** | | | |
| Male | 0.897 (0.708–1.137) | | |
| Female | 0.880 (0.657–1.178) | | 0.272 |
| **BMI** | | | |
| <24kg/m$^2$ | 0.974 (0.698–1.359) | | |
| ≥24kg/m$^2$ | 0.777 (0.620–0.975) | | 0.912 |
| **Smoking status** | | | |
| Never | 0.771 (0.598–0.993) | | |
| Former | 0.882 (0.522–1.491) | | 0.965 |
| Current | 1.031 (0.747–1.422) | | |
| **Hypertension** | | | |
| Yes | 0.792 (0.641–0.979) | | |
| No | 0.795 (0.525–1.204) | | 0.632 |
| **Dyslipidemia** | | | |
| Yes | 0.932 (0.691–1.257) | | |
| No | 0.681 (0.534–0.869) | | 0.570 |
| **Diabetes** | | | |
| Yes | 0.867 (0.662–1.136) | | |
| No | 0.731 (0.569–0.937) | | 0.497 |
| **History of stroke** | | | |
| Yes | 0.578 (0.319–1.048) | | |
| No | 0.802 (0.658–0.978) | | 0.778 |
| **Statins application history** | | | |
| Yes | 1.064 (0.746–1.519) | | |
| No | 0.700 (0.552–0.888) | | 0.021 |

Odds ratio (95%CI)

**Figure 2 Stratified analysis of the association (odds ratio (OR) (95% CI)) between the serum IgM level and the severity of coronary artery lesion.** Values are adjusted for age, gender, smoking status, obesity or overweight, hypertension, dyslipidemia, diabetes, stroke, and statins application history, stratifying factors excepted. $P < 0.05$ is considered statistically significant.

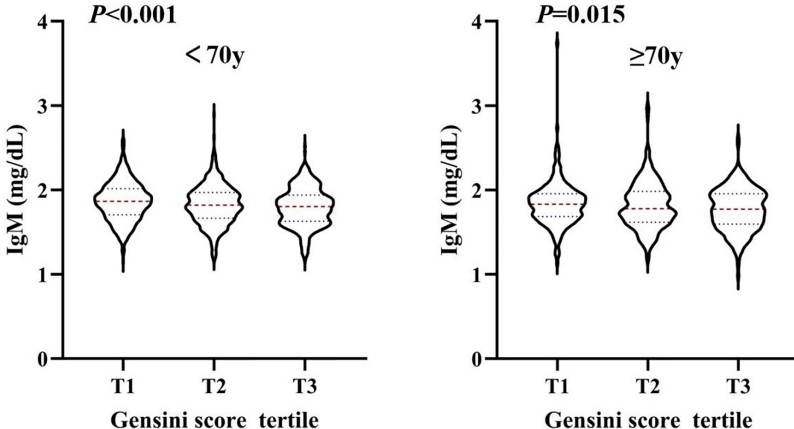

**Figure 3 The level of serum IgM in people younger than 70 years old and older than 70 years old, respectively, according to Gensini score tertile.** Violin plots of serum IgM concentrations in people younger than 70 years old (A) and older than 70 years old (B) by Gensini score tertile, showing median (red dashed line) and interquartile ranges (blue dashed line) on a log10 scale. $P < 0.05$ is considered statistically significant.

smoked, had hypertension, did not have hyperlipidaemia or diabetes, had no history of stroke, and did not have a history of statin use. In addition, we observed that the association between IgM levels and CAD varied by age and statin use history ($P_{interaction}$ = 0.029, $P_{interaction}$ = 0.021). After adjusting for sex, BMI, smoking status, hypertension, dyslipidaemia, DM, history of stroke, and statin use history, we found that IgM levels were associated with CAD in people younger than 70 years of age (OR: 0.730, 95% CI [0.579–0.921], $P$ = 0.008), but not in people older than 70 years of age (OR: 0.835, 95% CI [0.608–1.147], $P$ = 0.266). Additionally, IgM levels were associated with CAD in people without a history of statin use (OR: 0.700, 95% CI [0.552–0.888], $P$ = 0.003), but not in people with a history of statin use (OR: 1.064, 95% CI [0.746–1.519], $P$ = 0.731) after adjusting for age, sex, BMI, smoking status, hypertension, dyslipidaemia, DM, and history of stroke (Fig. 2). We further analysed the relationship between IgM levels and CAD severity according to the Gensini score. IgM levels exhibited a negative association with the Gensini score across different groups: patients younger than 70 years of age, patients older than 70 years of age (Fig. 3), and patients both with and without a history of statin use (Fig. 4).

## DISCUSSION

In this study, we observed that serum IgM concentration was associated with the traditional risk factors and severity of CAD. IgM levels gradually decreased with an increase in both the number of obstructed vessels and the Gensini score. After adjusting for age and sex in the logistic regression models, higher IgM levels exhibited a significant negative association with the severity of CAD, as assessed by the Gensini score. The relationship was still significant after further adjusting for smoking status, hypertension, dyslipidaemia, diabetes, stroke, and statin use history.

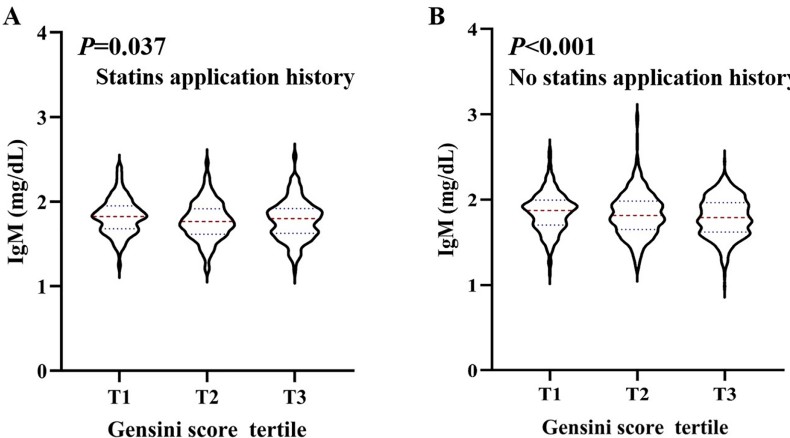

**Figure 4 The level of serum IgM in people with statins application history and no statins application history, respectively, according to Gensini score tertile.** Violin plots of serum IgM concentrations in people with statins application history (A), and no statins application history (B), by Gensini score tertile, showing median (red dashed line) and interquartile ranges (blue dashed line) on a log10 scale. $P < 0.05$ is considered statistically significant.

We investigated whether IgM levels were associated with CAD in a Chinese coronary angiography cohort. First, we analysed the relationship between serum IgM concentrations and traditional CAD risk factors. The results revealed a strong association between IgM levels and the concentrations of TC, TG, and LDL-C, as well as a positive association between IgM concentration and TP, HDL-C and ALB. Both high TC and high LDL-C levels play an important role in CAD (*Mostaza et al., 2000*; *O'Donoghue et al., 2014*). We found positive associations between IgM and TG, TC and LDL-C (Table 2), which could possibly be related to statin use. Logistic regression analysis showed that dyslipidaemia was not associated with CAD severity. The relationship between the clinical diagnosis of dyslipidaemia and CAD was not significant due to statin use.

Atherosclerosis is the primary cause of cardiovascular deaths related to heart attacks and strokes (*Kyaw et al., 2011*). It is considered to be a lipid-driven chronic inflammatory disease that begins with the accumulation of translocated lipids in arterial intimal layers. Recent studies have found that plaque development, progression, and rupture are modulated by the immune system (*Kyaw et al., 2012*). The immune inflammatory response is considered to be the most important pro-atherosclerotic factor. Although total serum immunoglobulins (Igs) are not normally considered relevant to clinical cardiovascular (CV) disease, there is substantial evidence in the preclinical literature suggesting that IgM is associated with atherosclerosis. The earliest evidence supporting an atheroprotective role of natural IgM is that sIgM−/− Ldlr−/− mice, which cannot secrete IgM, develop significantly more extensive atherosclerosis than Ldlr−/− controls (*Lewis et al., 2009*). Most serum IgM is natural IgM, which is produced spontaneously by a distinct subset of B cells and does not require any stimulation by foreign antigens or microorganisms (*Nguyen & Baumgarth, 2016*). Studies have found that natural IgM exerts a protective effect by interfering with the interaction between ox-LDL and macrophages, thereby preventing the formation of foam cells and limiting the proinflammatory effects of

ox-LDL (*Binder et al., 2003*; *Hörkkö et al., 1999*). Likewise, natural IgM promotes IgM deposits and reduces necrotic core size in atherosclerotic lesions, suggesting that IgM Abs has a protective effect against atherosclerosis (*Kyaw et al., 2011*). In addition, natural IgM may enhance the clearance of oxidized phospholipid-bearing apoptotic cells that accumulate in atherosclerotic plaques (*Thorp et al., 2008*). Serum IgM also contains a small amount of antigen-specific IgM produced by B1b cells when they are activated by both nonantigenic and antigen-dependent stimuli. *Rosenfeld et al. (2015)* recently found that B1b cells could also produce atheroprotective OSE-reactive IgM antibodies, which protect against atherosclerosis in mice. This finding suggests that similar mechanisms may occur in humans (*Rosenfeld et al., 2015*).

Several studies on human subjects have shown that serum IgM antibodies to oxidation-specific epitopes (OSEs) are inversely correlated with CAD. For example, serum IgM antibodies to copper-oxidized low-density lipoprotein (CuOx-LDL) and malondialdehyde-modified low-density lipoprotein (MDA-LDL) are inversely correlated with carotid intima media thickness and the risk of developing a >50%-diameter stenosis in the coronary arteries (*Karvonen et al., 2003*; *Tsimikas et al., 2007*, *2012*). Similarly, IgM titres to OSE phosphocholine have been reported to be inversely correlated with the incidence of heart attack and CAD risk in patients with lupus (*Anania et al., 2010*; *Grönlund et al., 2009*; *Grönwall et al., 2012*). In addition, serum IgM is negatively associated with baPWV in women and negativelya ssociated with a lower risk of arterial stiffness, which is also an index of subclinical atherosclerosis (*Liu et al., 2019*). Inflammation and autoimmune responses play important roles in CAD, yet little is known about whether serum IgM levels are reduced in CAD.

Stratified analysis results showed that IgM levels were associated with CAD severity based on the number of affected regions, affected vessels, and Gensini score tertiles (Fig. 1). The Gensini score was used to evaluate the severity of coronary artery disease. Because the Gensini score included different percentages of stenosis, it was more indicative of the degree of atherosclerosis than the Syntax score. *Khamis et al. (2016)* also found that total serum IgM levels were lower in patients with CAD and stroke than in controls, which is consistent with our results. In addition, logistic regression analysis revealed that after adjusting for traditional risk factors in CAD patients, serum IgM levels were still negatively and independently associated with the Gensini score. Additional stratified analyses showed a strong association between IgM levels and CAD in people younger than 70 years old and in those without a history of statin use (Fig. 2). These results indicated that IgM may be involved in the progression of atherosclerosis, especially in people younger than 70 years old and those without a history of statin use. *Palmieri et al. (2021)* found that IgM antibodies against MDA bound to human albumin were significantly decreased in 60-year-olds who developed CAD within a 5-year follow-up, linking IgM anti-MDA levels above the 66th percentile to a reduced risk of CAD. This was independent of other traditional risk markers. Our study also demonstrated that serum IgM concentrations decreased with age, and the protective effect of IgM on atherosclerosis was particularly pronounced in people younger than 70 years old. We propose that CAD is independently associated with serum IgM in the coronary angiography population, which may enable IgM to serve as a

biomarker for the severity of CAD. Existing studies generally report that IgM has a protective effect against atherosclerosis. IgM may act by neutralizing the proinflammatory properties of oxidized low-density lipoprotein (ox-LDL), inhibiting the uptake of ox-LDL by macrophages, and promoting apoptotic cell clearance (*Rosenfeld et al., 2015*; *Tsiantoulas et al., 2014*). Additionally, the role of specific antibodies as biomarkers of atherosclerosis has been extensively studied, especially antibodies to epitopes induced by oxidative modification of low-density lipoprotein (LDL). *Tsimikas et al. (2012)* found that IgG Cu-OxLDLs were associated with a higher risk of CVD, while IgM MDA-LDLs were associated with a lower risk of CVD. In addition, *Khamis et al. (2016)* found that while IgM anti-MDA-LDL antibodies were significantly associated with freedom from CV events, these antibodies were also significantly associated with serum IgM levels. Therefore, these researchers suggested that in hypertensive patients, total serum IgM levels are independent predictors of the absence of CAD in general. Total serum Ig levels significantly improve the risk classification of cardiovascular events (*Khamis et al., 2016*). Thus, comparing only one form of IgM anti-MDA levels may overlook the predictive power of other specific antibodies. In fact, total serum IgM is a more favourable biological indicator.

Our study still has some limitations. First, as this was a cross-sectional study, it was difficult to determine the causal association between IgM and CAD severity in Chinese patients, so our findings need to be confirmed in further prospective studies. Second, the control subjects who had a <50% narrowing of coronary arteries were not truly healthy individuals, which may have caused an underestimation of the association between IgM and CAD and limited the power of this study.

## CONCLUSION

In conclusion, we analysed the relationships between serum IgM levels and CAD severity in Chinese patients. The results showed that reduced IgM levels were associated with a higher severity of CAD, especially in people younger than 70 years old and those without a history of statin use.

### Funding

This study was supported by the National High Level Hospital Clinical Research Funding (Grant No BJ-2022-124 and No BJ-2022-113) and the CAMS Innovation Fund for Medical Sciences (No. 2021-I2M-1-050). The funders had no role in study design, data collection and analysis, decision to publish, or preparation of the manuscript.

### Grant Disclosures

The following grant information was disclosed by the authors:
National High Level Hospital Clinical Research Funding: BJ-2022-124 and No BJ-2022-113.
CAMS Innovation Fund for Medical Sciences: 2021-I2M-1-050.

## Competing Interests

The authors declare that they have no competing interests.

## Author Contributions

- Yanan Zhang performed the experiments, analyzed the data, prepared figures and/or tables, and approved the final draft.
- Xi Qi analyzed the data, prepared figures and/or tables, and approved the final draft.
- Siming Wang conceived and designed the experiments, authored or reviewed drafts of the article, and approved the final draft.
- Wenduo Zhang performed the experiments, authored or reviewed drafts of the article, and approved the final draft.
- Ruiyue Yang conceived and designed the experiments, authored or reviewed drafts of the article, and approved the final draft.
- Xinyue Wang conceived and designed the experiments, performed the experiments, authored or reviewed drafts of the article, and approved the final draft.
- Wenxiang Chen performed the experiments, prepared figures and/or tables, and approved the final draft.
- Fusui Ji conceived and designed the experiments, prepared figures and/or tables, and approved the final draft.
- Jun Dong conceived and designed the experiments, authored or reviewed drafts of the article, and approved the final draft.
- Xue Yu conceived and designed the experiments, performed the experiments, analyzed the data, authored or reviewed drafts of the article, and approved the final draft.

## Human Ethics

The following information was supplied relating to ethical approvals (*i.e.*, approving body and any reference numbers):

Beijing Hospital Ethics Committee granted ethical approval to conduct the study (2016BJYYEC-121-02).

## Data Availability

The raw data are available in the Supplemental File.

## Supplemental Information

Supplemental information for this article can be found online at http://dx.doi.org/10.7717/peerj.17012#supplemental-information.

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
