# Peer review of "Serum immunoglobulin M is associated with the severity of coronary artery disease in adults"

_PeerJ, doi:10.7717/peerj.17012_

## Round 0.1 · original submission · Major Revisions

In this paper, the authors evaluate the relationship between total serum immunoglobulin M (IgM) and the severity of coronary artery disease (CAD) in Chinese patients who underwent coronary angiography. The authors analyzed more than 2,000 patients and concluded that serum IgM was independently negatively associated with the severity of coronary artery disease diagnosed by angiography in Chinese adults. Despite recognizing the importance of the study, several criticisms concerning the data analyses and interpretations may be considered.

The manuscript does not provide details on how the levels of IgM in the patient's serum were measured. As pointed out by the reviewers, the statistical analyses and experimental design must be further clarified.

The absence of data about targets recognized by IgM prevents progress in the exploratory discussion of potential mechanisms involved in the protection of this subclass of antibodies against atherosclerosis. As noted by the authors, other studies have analyzed the IgM profile against specific molecules, such as phosphorylcholine, copper-oxidized low-density lipoprotein (CuOx-LDL), and malondialdehyde-modified low-density lipoprotein (MDA-LDL). It is necessary to better justify the use of a single immunological parameter to explain the association between total IgM levels and CAD and critically consider the study's limitations.

The English language also needs to be improved to make the manuscript clear to readers.

Reviewer 1 ·

Basic reporting

In the submitted manuscript by Zhang et al., the authors investigated the relationship between serum immunoglobulin M (IgM) and the severity of coronary artery disease in Chinese patients who underwent coronary angiography. They discovered the negative correlation between IgM level and the severity of CAD patients.
The English language can improved for this manuscript.
Could the authors show the full name for NCAD which is not indicated in the context.

Experimental design

1. The limitation of the studies is that only one parameter is analyzed that is the level of IgM in the serum. How about the about immunoglobulin like IgA, IgG etc. And as B cells are the main cell that can produce IgM, is there dfference in B cell proportion between CAD and NCAD?
2. The authors mentioned many times that they adjust the data for different parameters like age and sex. Could the authors be more specific on how the adjustment is performed?
3. In the results, they used the word ’significant association’ which is ambiguous. Please be more specific like negative or positive.
4. The p value indicated in the figure is not precise. The authors do not show which group that are comparing.

Validity of the findings

N/A

Reviewer 2 ·

Basic reporting

In lines 84-86, you mentioned that “this study, the Beijing Hospital Atherosclerosis Study .. was a prospective study”. In line 87, you mentioned that “our study is a retrospective study”. In line "290", you mentioned that this was a "cross-sectional study". These are three different study designs. Please change your wording to make it more clear what your study design is.

Experimental design

- In table 1, "Comparison of baseline characteristics of study population according to serum
IgM tertile", could you describe what statistical tests you used for continuous variables and categorical variables respectively? You have mentioned in line 128 that the Mann-Whitney U test was used to compare continuous variables between groups. The Mann-Whitney test can only be used to compare two groups. To compare three or more groups, you should the Kruskal-Wallis test instead. Also, could you describe how you obtained the p-value for trend? I don’t think you can obtain the p-value for trend from either the Mann-Whitney test or the chi-square test since both of them treat the IgM tertile as a nominal variable instead of a ordinal variable.
- Could you provide supplementary tables to show the ORs and 95% CIs of all the variables included in the logistic regression models that were used for testing interactions? It will show the reader how you defined the interaction term if you included the two variables involved in the interaction term as main effects, and what other variables were included in the same model,
- Could you provide the R_sqaured, and the F-statistics P-value for each regression model? This could help show whether your logistic regression models are statistically significant and whether the interpretations based on the logistic regression models are valid.
- Do you have any missing data in your study? Any participants failed to respond to any questions on the survey? If not, please indicate this in your result section. If you do have missing data, in table 2, could you show the number of samples included in each correlation coefficient test? In table 3, could you show the number of samples included in each logistic regression model ?

Validity of the findings

no comment

---

## Round 0.2 · accepted · Accept

The authors have satisfactorily responded to all comments and made the necessary changes to the manuscript.

Reviewer 1 ·

Basic reporting

N/A

Experimental design

N/A

Validity of the findings

N/A

Additional comments

I appreciate the authors' detailed point-to-point response to my previous comments. I look forward to the final version of the manuscript. Congrats!

Reviewer 2 ·

Basic reporting

The authors have adequately addressed my comments. Therefore, I have no further comments.

Experimental design

The authors have adequately addressed my comments. Therefore, I have no further comments.

Validity of the findings

The authors have adequately addressed my comments. Therefore, I have no further comments.